# Co-Occurrence of Overweight/Obesity, Anemia and Micronutrient Deficiencies among Non-Pregnant Women of Reproductive Age in Ghana: Results from a Nationally Representative Survey

**DOI:** 10.3390/nu14071427

**Published:** 2022-03-29

**Authors:** Aaron K. Christian, Matilda Steiner-Asiedu, Helena J. Bentil, Fabian Rohner, Rita Wegmüller, Nicolai Petry, James P. Wirth, William E. S. Donkor, Esi F. Amoaful, Seth Adu-Afarwuah

**Affiliations:** 1Regional Institute for Population Studies, University of Ghana, Legon P.O. Box LG 96, Ghana; akchristian@ug.edu.gh; 2Department of Nutrition and Food Science, University of Ghana, Legon P.O. Box LG 134, Ghana; tillysteiner@gmail.com; 3Department of Nutrition and Food Sciences, University of Rhode Island, Kingston, RI 02881, USA; helena_bentil@uri.edu; 4GroundWork, 7036 Fläsch, Switzerland; fabian@groundworkhealth.org (F.R.); rita@groundworkhealth.org (R.W.); nico@groundworkhealth.org (N.P.); james@groundworkhealth.org (J.P.W.); william@groundworkhealth.org (W.E.S.D.); 5Nutrition Department, Ghana Health Service, Accra P.O. Box MB 582, Ghana; esiforiwa@gmail.com

**Keywords:** overweight, obesity, double burden of malnutrition, micronutrient deficiencies, women of reproductive age, anemia

## Abstract

Overweight/obesity (OWOB) often co-occurs with anemia or micronutrient deficiencies (MNDs) among women of reproductive age (WRA) in Ghana; identifying the risk factors of these conditions is essential for prevention. We aimed to examine the prevalence of OWOB, anemia, and MNDs and their co-occurrence and risk factors among non-pregnant women 15–49 years of age in Ghana. Data were from a 2017 two-stage national survey of 1063 women. We estimated the weighted prevalence of single and co-occurring malnutrition, and used logistic regression to explore risk factors. The prevalence of OWOB, anemia, and ≥1 MND was 39%, 22%, and 62%, respectively; that of OWOB co-occurring with anemia was 6.7%, and OWOB co-occurring with ≥1 MND was 23.6%. There was no significant difference between observed and expected prevalence of co-occurrence OWOB with anemia or MND. Risk factors were: living in southern (vs. northern) belt, high- (vs. low-) wealth household, being ≥ 25 years old, and being married (vs. single) for OWOB, and living in northern (vs. southern) belt and medium- (vs. low-) wealth household for anemia and ≥1 MND, respectively. Different interventions are required for addressing OWOB in WRA than those for anemia and MNDs.

## 1. Introduction

For decades, research on nutrition in low-and middle-income countries (LMICs) focused largely on the burden of undernutrition in vulnerable groups [1,2,3]. The increasing prevalence of individuals that are overweight or with obesity (OWOB) among women of reproductive age (WRA; 15–49 years of age) in many LMICs represents a major public health concern; OWOB is directly linked with non-communicable diseases and also poses a serious threat to maternal mortality due to its association with obstetric complications and subsequently, infant mortality [4,5]. Factors contributing to the current increase in OWOB in LMICs include urbanization and associated epidemiological and nutrition transitions, which create obesogenic food environments [6,7]. While there is an overall decrease in the prevalence of micronutrient deficiencies and anemia, there are women caught up in the middle of the nutrition transition who are OWOB and at the same time anemic or with a micronutrient deficiency [8,9].

The 2020 *Lancet Series* on the Double Burden of Malnutrition (DBM) indicates that there has been a slow response by the global health community to reducing the DBM in developing countries [10]. Prior to the *Lancet Series,* other authors argued that there is a need for a paradigm shift from the narrow focus on undernutrition within the framework of the Millennium Development Goals to a wider attention on the double burden of malnutrition as captured in the Sustainable Development Goals [11]. Understanding country-specific social and biological pathways to the DBM, particularly among WRA, will lead to appropriate and relevant interventions [12].

Ghana, like other LMICs, is not exempt from the current OWOB epidemic [13,14]. Data from Ghana’s nationally-representative Demographic and Health Surveys (GDHS) reveals a steady increase in the prevalence of OWOB (overnutrition) among women aged 15–49 years from 25% to 40% between 2003 and 2014 [15]. This notwithstanding anemia prevalence in the same period fluctuated between 45% and 59% [16]. Based on these results from the GDHS, Ghanaian WRA are classified as having a high prevalence of OWOB and of anemia, as the national prevalence for these conditions exceeds the World Health Organization (WHO) threshold of 20% for overweight [17] and 40% for anemia [18].

Few studies in Ghana, however, have examined household and individual factors that may influence the presence of the double burden of malnutrition in individuals. For example, the study by Kushitor et al. (2020) examined the correlates of the double burden of malnutrition among women, and the analysis was limited to OWOB and anemia [19]. Women’s age, marital status, wealth, and parity were identified as the risk factors for the examined double burden of malnutrition. The non-nutritional factors such as malaria, inflammation and hemoglobinopathies might cause anemia in sub-Saharan Africa (SSA) [20], and using anemia as a proxy for micronutrient deficiency could result in inaccurate estimates of specific micronutrient deficiencies. In the present study, we aimed to determine the prevalence and risk factors of OWOB coexisting with (i) anemia and (ii) micronutrient deficiencies among WRA in Ghana.

## 2. Materials and Methods

### 2.1. Study Design

This study used data from the 2017 Ghana Micronutrient Survey (GMS) 2017 [21]. The GMS was a national survey examining micronutrient status indicators among women from the three strata (Southern, Middle and Northern Belt) in Ghana. The number of households sampled from each strata was based on differences in household sizes across the regions [16]. The survey was conducted as a two-stage cluster sample survey. Based on Ghana’s 2010 national census, 30 enumeration areas (EAs) were selected in each stratum [22]. This was followed by an updated listing of all households in each EA and then a random selection of households with equal probability of selection. All non-pregnant WRA from 50% of randomly selected households were recruited.

Questionnaires were written in English, however, interviews were conducted in the preferred language(s) of the respondents, mostly Twi, Ga, Fante, Ewe in the Southern and Middle strata and Dagaare, Waale, Frafra, and Dagbani in the Northern belt. Household interviews were used to collect information on household composition, ownership of durable goods or other assests, and education and sex of the household head. We used interviews with women to collect information on age, educational status, and recent supplement consumption. In addition, the Minimum Dietary Diversity for Women (MDD-W) questionnaire module was also administered to each woman so that dietary diversity could be calculated [23]. The study protocol was approved by the Ethics Review Committee of the Ghana Health Service (GHS-ERC: 15 January 2017) and was registered on the Open Science Framework platform (https://osf.io/j7bp9/ (accessed on 22 February 2022)). Written informed consent was obtained from all respondents. 

A total of 2123 households consented for the GMS 2017 survey, and 1063 non-pregnant WRA were recruited from 50% of the households. Approximately 1% of this sample refused or was unable to participate in the interview and 91% consented to blood collection. For folate and vitamin B12 status assessment, a random sub-sample of 50% of the enrolled women was analyzed. The final sample size for assessing the co-occurrence of OWOB and anemia was 989 and that of the co-occurrence of OWOB and any micronutrient deficiency was 457.

### 2.2. Anthropometric Measurements, Blood Collection and Laboratory Analysis

Upon completion of the interviews, non-pregnant WRA were invited to a central location within each cluster, where trained anthropometrists used standard procedures [24] to measure the women’s weight to the nearest 100 g (Seca scale model 877, Hamburg, Germany) and height to the nearest 0.1 cm (height board, UNICEF item number S0114540, Copenhagen, Denmark). Subsequently, trained phlebotomists collected 6 mL of venous blood into silica-coated serum tubes (Becton Dickinson Vacutainer, Franklin Lakes, NJ, USA).

A small portion of the blood was extracted from the blood tubes to determine hemoglobin concentration by means of a portable hemoglobinometer (Hb301, HemoCue AB, Ängelholm, Sweden), and to determine malaria parasitemia by means of the SD BIOLINE Malaria Ag Pf/Pan rapid diagnostic test kit (Standard Diagnostics Inc, Gyeonggi-do, Republic of Korea). The remaining blood samples were centrifuged at 3000 rpm for 10 min. The serum aliquots were frozen at −20 °C until being air-freighted with dry ice for analysis of retinol-binding protein (RBP), ferritin, C-reactive protein (CRP), and α1-acid glycoprotein (AGP) at the VitMin-Lab (Willstaett, Germany) using a sandwich ELISA. Folate and vitamin B12 were analyzed in a 50% sub-sample of women at the USDA/ARS Western Human Nutrition Research Center (Davis, CA, USA) using the Cobas e411 analyzer (Roche Diagnostics, Indianapolis, IN, USA).

### 2.3. Case Definitions

Women’s body mass index (BMI) values (kg/m^2^) were categorized as follows: 24.9 and below as underweight/normal, 25.0–30.0 overweight, and >30.0 respondent with obesity [25]. We defined iron deficiency as inflammation-adjusted serum ferritin <15 μg/L [26,27], vitamin A deficiency as RBP < 0.7 μmol/L [28], folate deficiency as serum folate <10 nmol/L, vitamin B-12 deficiency as serum B12 concentrations <148 pmol/L, and any anemia as Hb < 120 g/L [18,29]. Household sanitation was considered improved if the sanitation facility was connected to a public sewer or septic system, or consisted of a pour-flush latrine, a ventilated improved pit latrine, or a simple pit latrine. Households using public water systems, boreholes, protected dug wells, protected springs, or rainwater as their source of drinking water were also considered to have an improved source of water [30]. 

### 2.4. Statistical Analysis

We performed the analysis using Stata statistical software package, Version 14.2 (2017; StataCorp, College Station, TX, USA). To account for the unequal probability of selection in the three strata, the current analysis was weighted to avoid inaccurate estimates of primary sampling unit size used during stage one sampling. We generated a household wealth index using the principal components analysis of household assets and dwelling materials (e.g., materials for roofs, outer walls, and flooring); wealth terciles were then calculated [31]. Proportions of the prevalence of OWOB, micronutrient deficiencies including anemia and the double burden of OWOB and micronutrient deficiencies were determined. We used Pearson’s chi-squared test to test differences in proportions. The product of the observed prevalence of OWOB and anemia as well as the product of the observed OWOB and any micronutrient deficiency were calculated as the expected frequencies of co-occurrence. The squared difference of the observed and expected frequencies of co-occurrence divided by the expected was the chi-squared statistic with one degree of freedom. We used multiple logistic regression to determine the predictors of OWOB, anemia and ≥1 micronutrient deficiency (iron, vitamin A, folate, or vitamin B-12). The margins command in stata (*margins,dydx(*)*) was used to estimate the results in Table 4 for easy interpretation. The variables considered as predictors were women’s age [32], education [33], marital status [34], household wealth index category [35], place of residence (urban or rural) [36], dietary diversity [37], and household access (yes or no) to improved sanitation and source of drinking water (safe or unsafe).

## 3. Results

### 3.1. Background Characteristics of the Respondents

The mean age of women in the overall sample was 29 years, and 74% of the women were partly or fully literate (Table 1). The highest percentage of the women were 15–24 years of age (38%) and were married (60%). The vast majority (90%) lived in households with safe drinking water but only 13% of women had access to improved sanitation in their households. Approximately half of the women lived in urban areas, and the middle belt stratum had the highest percentage (45%) of the total number of households. While over 60% of the women had a BMI below 25, about a quarter were overweight and 14% were respondents with obesity. The mean hemoglobin concentration was 128 g/L and nearly 22% of the women were anemic (Figure 1). Vitamin A deficiency had the lowest prevalence (1.5%) and the most prevalent micronutrient deficiency was that of folate (54%).

Table 2 shows the results on the nutritional status among women by select household and sociodemographic status. More than half (66%) of the women had at least one micronutrient deficiency. The prevalence of OWOB among the women was highest in the Southern Belt (47%) compared to the Middle (41%) and Northern (19%) Belts (*p* < 0.001), and higher in urban areas (47%) than in the rural (29%) (*p* < 0.001). The prevalence of OWOB was higher in women in high wealth households (57%) compared to those in low wealth (22%) households (*p* < 0.001) and higher in women belonging to households with an improved water source (44%) compared with those with an unimproved (38%) water source (*p* < 0.001). The prevalence of OWOB increased with age. A higher proportion (57%) of older women (35–49 years) were OWOB compared to the younger (15–24 years) women (16%) (*p* < 0.001).

The prevalence of anemia was lowest in the Middle Belt (17%) compared to the Southern (24%) and Northern (28%) Belt (*p* < 0.030) The prevalence of having at least one micronutrient deficiency was lower in older women (50%) compared to younger women (67%) *p* < 0.013. 

The prevalence of the co-occurrence of OWOB and anemia and the co-occurrence of OWOB and ≥1 micronutrient deficiency was 7% and 24%, respectively. The prevalence of co-occurrence of OWOB with anemia was significantly lower in women in low wealth households (1%) compared to their counterparts in other wealth categories (10% in medium and 8% in high: *p* < 0.001) and higher in women from households with improved sanitation (unimproved 6% vs. 13% improved, *p* = 0.001).

The prevalence of co-occurrence of OWOB with anemia was highest (10%) in 25–35-year-old women (3% in 15–24 years and 8% in 35–49 year old women, *p* < 0.001). The prevalence of co-occurrence of OWOB with at least one micronutrient deficiency was highest in the Southern Belt (28% vs. 25% and 12% in the Middle Belt and Northern Belt, respectively, *p* < 0.001), and higher in urban areas (30% vs. 18%, *p* < 0.001) than in the rural areas. A higher proportion of wealthier women (36% in high to 13% in low wealth quintile, *p* < 0.001) were OWOB and had at least one micronutrient deficiency. The prevalence of co-occurrence of OWOB with at least one micronutrient deficiency was highest (31%) in 25–34-year-old women (13% in 15–24 and 29% in 35–49-year-old women, *p* < 0.001).

### 3.2. Co-Occurrence of OWOB and Anemia and OWOB with Micronutrient Deficiency

The expected frequencies of co-occurrence of OWOB and anemia and co-occurrence of OWOB and ≥1 micronutrient deficiency and were 9% and 24% respectively. There was no statistically significant difference between the observed and expected frequency of co-occurrence nor did these comparisons vary by stratum (Table 3).

### 3.3. Predictors of OWOB, Anemia, and Micronutrient Deficiency

Location of residence, household wealth, living in a household with improved sanitation, being aged 25–49 years, and being married/cohabiting were independent predictors of OWOB (Table 4). Specifically, women living in the northern zone were 11% less likely to be OWOB when compared to women living in the southern zone. Women belonging to medium and high-wealth households were 8% and 21% more likely to be OWOB when compared to their counterparts living in low wealth households. While older women (25–49 years old) were 30% more likely to be OWOB than young women (15–24 years old), married women or women cohabiting with their partners were also 10% more likely to be OWOB than single women. Women in the middle zone were 8% less likely and older women were 7% less likely to be anemic compared to women in the southern zone and younger women (15–24 year), respectively. Women living in the northern zone were 20% more likely to suffer from at least one micronutrient deficiency (not anemia) when compared to women living in the southern zone, while women belonging to medium wealth households were 16% more likely to have at least one micronutrient deficiency compared to women in low wealth households.

## 4. Discussion

Similar to an earlier study conducted in Malawi [40,41], there was no meaningful difference in the observed co-occurrence of OWOB and anemia as well as the co-occurrence of OWOB and micronutrient deficiency and their expected prevalence, suggesting that OWOB, anemia, and micronutrient deficiencies occur mostly independently. This result suggests that the co-occurrence of these malnutrition conditions may be a result of chance rather than having some shared determinants. Thus, we will discuss the correlates of OWOB, anemia and micronutrient deficiency separately first and then, discuss the DBM findings in more detail.

### 4.1. Overweight/Obesity

The prevalence of OWOB (39%) observed by the GMS in 2017 shows that there was little change since 2014, when the DHS estimated that 40% of women were OWOB [16]. The high prevalence can be attributed to the current nutrition transition, which is characterized by an increase in obesogenic diets with less diversity and a trend towards consumption of less nutrient dense foods coupled with a decrease in physical activity [42]. For example, it has been shown that exposure to television (ownership and viewing) and being less physically active are associated with OWOB among Ghanaian women [43]. Another factor associated with women being OWOB in Ghana is the cultural preference for more plump body sizes in some Ghanaian traditional settings [44,45].

Women in the northern stratum were less likely to be OWOB compared to those in the southern. This could be due to a higher proportion of women in the northern region engaging in agriculture and also being more physically active than their southern counterparts. In their systematic review of overweight and obesity in Ghana, Ofori-Asenso et al. concluded that the observed pattern may partly be explained by the extent of urbanization observed in the different regions [46]. Generally, with urbanization comes an increased access to energy-dense foods and less strenuous jobs, which leads to many more people having a positive energy balance and hence becoming OWOB [47]. The northern region of Ghana is less urbanized making the population less vulnerable to some obesogenic environments like the presence of fast-food joints. Closely related to the overweight/obesity and urbanization discourse is that of wealth and overweight/obesity. Findings in our study showed that women in richer households were more likely to be OWOB. This result corroborates other researchers that found that while overweight/obesity is on the rise among the poor, the wealthy are still the most affected in some developing countries like Ghana [48]. Wealthier households can afford the more energy-dense foods, eat in fast food restaurants and are less engaged in strenuous jobs, thus less physically active [49]. Respondents from a study in some urban communities in Ghana perceived having a large body size as depicting an individual who eats well and is affluent, belonging to a high social class [50]. This is however contrary to more developed economies where more overweight/obese individuals are found among low socioeconomic households [51].

### 4.2. Anemia and Micronutrient Deficiency

Similar to overweight/obesity, the prevalence of anemia continues to be of public health significance in Ghana. The GDHS reported the prevalence of anemia among women (15–49 years) was 45% in 2003, 59% in 2008 and 42% in 2014. An earlier study explained that the observed difference in the prevalence of anemia reported in this study and that of the GDHS could be attributed to the use of different anemia testing devices and the different seasons in which the two studies were conducted, particularly the influence of malaria on anemia during the rainy season, which serves as a catalyst for increased vector (mosquito) breeding for the malaria parasite [21]. Women in the middle zone were found to be more likely to be anemic compared to those living in the southern zone. Given the multiple causes of anemia (inflammations due to infections, low dietary intake of iron and other micronutrients, inherited blood disorders) in Ghana, context-specific determinants of anemia are required to guide appropriate interventions. For instance, Petry et al. showed that while iron deficiency was associated with children being anemic in the Northern and Middle belts of Ghana, there was no such relationship among children in the Southern Belt [52]. Lopes et al. suggested in their systematic review of nutrition-specific interventions for preventing and controlling anemia that besides the continuous efforts to improve dietary diversity and quality, daily iron supplementation may increase the hemoglobin levels and decrease the risk of anemia in both non-pregnant and pregnant women.

Folate deficiency was the most prevalent micronutrient deficiency (54%) in this study, however, it is lower than that reported in other West African countries such as Côte d’Ivoire (86%) and Sierra Leone (79%) [53]. Generally, populations that consume more cereal staples and less green leafy vegetables and legumes are more likely to be folate deficient [54]. The high prevalence of folate deficiency in our study is not surprising as cereals and starchy staples are the most frequently consumed foods across all age groups in Ghana and other SSA countries [55]. Given the fact that serum folate level contributes to the risk of congenital birth defects, checking folate status could be recommended for both pre-pregnant and expecting mothers.

### 4.3. Double Burden of Malnutrition

With respect to the double burden of malnutrition, the prevalence of the co-occurrence of OWOB and anemia in Ghana (7%) was higher than that observed in other Sub-Saharan African (SSA) countries like Malawi (3%) and Ethiopia (2%), but lower than in other LMICs such as India (9%), Brazil (14%) and Guatemala (12%) [56,57]. The co-occurrence of OWOB with ≥1 micronutrient deficiency largely consisted of the prevalence of OWOB with folate deficiency, as the prevalence of the other measured micronutrients (iron deficiency, vitamin B-12 and vitamin A) was low. The co-occurrence of OWOB and micronutrient deficiencies has been found in other developing countries to be largely due to zinc deficiency. Zinc deficiency was however not examined in the current study [40]. The higher prevalence of the co-occurrence of OWOB and micronutrient deficiencies than OWOB and anemia was consistent with another nationally representative analysis [41]. The significantly higher proportion of co-occurrence of OWOB and micronutrient deficiency in urban areas is consistent with others [40]. Concurrently, there is cross-sectional research revealing that urban women in Sub-Saharan Africa do have inadequate intakes of micronutrients such as vitamin B12 and folate compared to their rural counterparts [58].

Evidence of higher co-occurrence of OWOB and anemia and OWOB with micronutrient deficiency with greater prevalence of OWOB in population-based surveys is limited [41]. Alternatively, there is evidence of metabolic interaction between being OWOB and anemic or micronutrient deficient due to the need for greater blood volume required by the increase in body weight, provoking micronutrient deficiencies and a decrease in the bioavailability of iron due to its sequestration in the reticuloendothelial system [59].

Given the burden of overweight/obesity, anemia, and micronutrient deficiencies in individuals and at the national level in developing countries, the co-occurrence of these malnutrition conditions poses aggravated stress on economies and well-being. This notwithstanding the absence of statistical difference between the observed prevalence and expected prevalence of the double burdened conditions suggests that OWOB, anemia, and micronutrient deficiencies should be treated more as independent conditions among non-pregnant women in Ghana. This notwithstanding that for individual women living with OWOB, weight loss reduction programs could lead to more efficient absorption of micronutrients (e.g., Fe) from the diet and hence impact their hemoglobin levels positively. Additionally, given that the incidence of OWOB and micronutrient deficiencies followed an inverse social gradient. Women in wealthier households were more likely to be OWOB and also more likely to be micronutrient deficient. This is consistent with previous studies showing a higher prevalence of overweight/obesity at the individual level among the most socio-economically advantaged, giving credence to the changing lifestyle that could be associated with this group.

A major strength of this study is the use of nationally representative data in assessing the prevalence of OWOB, anemia and other micronutrient deficiencies concurrently. Our findings suggest a multi-pronged approach that targets the various determinants of OWOB, anemia and micronutrient deficiency may be needed, and that individual-level DBM among non-pregnant women of reproductive age may not be prominently present.

## Figures and Tables

**Figure 1 nutrients-14-01427-f001:**
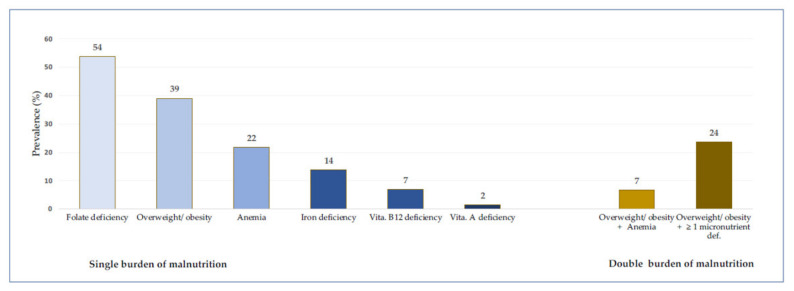
Prevalence of overweight/obesity (OWOB), anemia, micronutrient deficiencies and the co-occurrence of OWOB and anemia (i.e. Overweight/obesity + Anemia) and OWOB and at least a micronutrient deficiency (Overweight/obesity + ≥ 1 micronutrient deficiency).

**Table 1 nutrients-14-01427-t001:** Socio-demographics, anthropometry, and micronutrient status among non-pregnant women of reproductive age, Ghana 2017.

			Total	
Characteristic		*n*	% or Mean ^a^	95% CI ^b^
Household (*n* = 1063)				
Stratum				
	Southern, %	330	36.1	(29.7, 43.0)
	Middle, %	429	45.3	(38.5, 52.3)
	Northern, %	304	18.6	(14.7, 23.3)
Place				
	Urban, %	474	49.9	(37.4, 62.5)
	Rural, %	589	50.1	(37.5, 62.6)
Improved Sanitation, %		124	13.0	(8.8, 18.8)
Safe drinking water, %		907	90.3	(84.4, 94.1)
Wealth				
	Low, %	429	31.4	(24.3, 39.4)
	Medium, %	341	37.3	(29.4, 45.9)
	High, %	293	31.4	(23.1, 41.0)
Women age				
	Mean age (years)	1060	29.1	(28.5, 29.8)
	15-to 24, %	406	38.2	(34.7, 41.9)
	25-to 34, %	371	35.2	(31.9, 38.6)
	35-to 49, %	283	26.6	(24.2, 29.0)
Woman’s literacy status				
	Illiterate, %	210	25.8	(21.7, 30.3)
	Partly or fully literate, %	590	74.2	(69.7, 78.3)
Marital status				
	Married, %	405	59.9	(56.5, 63.2)
	Unmarried, %	647	40.1	(36.8, 43.5)
*Anthropometry*				
	Body Mass Index (kg/cm^2^), mean	1002	24.5	(24.1, 25.0)
	Underweight/Normal weight, % ^c^	641	61.0	(56.3, 65.5)
	Overweight, %	232	24.7	(21.0, 28.8)
	Obesity, %	129	14.3	(11.5, 17.7)
*Micronutrient status*				
Hemoglobin concentration				
	Hemoglobin (g/L), mean	999	127.7	(126.4, 128.9)
	Any anemia, %	999	21.7	(18.7, 25.1)
Iron status (*n* = 987)				
	Ferritin (μg/L), median ^d^ (IQR)	987	43.1	(23.3, 72.4)
	Iron deficiency, % ^e^		13.7	(11.2, 16.6)
Vitamin A status (*n* = 987)				
	RBP (μmol/L), mean ^f^	987	1.6	(1.6, 1.7)
	Vitamin A deficiency, % ^g^	987	1.5	(0.8, 2.9)
Folate status (*n* = 473)				
	Serum folate, median (IQR)	473	9.3	(5.4, 16.4)
	Folate deficiency, % ^h^	473	53.8	(47.6, 60.0)
Vitamin B12 status (*n* = 471)				
	Serum vitamin B12 (pmol), mean	471	454.0	(426.8; 481.3)
	Vitamin B12 deficiencyi ^i^, %	471	6.9	(4.8, 9.8)

^a^ Percentages/means weighted for unequal probability of selection. ^b^ CI = confidence interval, calculated considering the complex sampling design. For median calculations, inter-quartile range is provided. ^c^ Underweight defined as BMI < 18.5; normal BMI defined as BMI 18.5–24.9; overweight defined as BMI 25.0–29.9; respondents with obesity defined as BMI > 30. ^d^ Adjusted for inflammation [38]; corresponding unadjusted median ferritin concentration 45.9 μg/L (IQR: 25.0; 83.7). ^e^ Based on inflammation-adjusted ferritin concentration [38]. ^f^ Retinol-binding protein, adjusted for inflammation [39]. ^g^ Based on the retinol-binding protein, adjusted for inflammation [39]. ^h^ Folate deficiency defined as serum folate <10 nmol/L; this analyte was measured only in a random sub-sample of women. ^i^ Vitamin B12 deficiency and marginal status defined as plasma B12 < 148 pmol/L and plasma B12 ≥ 148 and < 220; this analyte was measured only in a random sub-sample of women.

**Table 2 nutrients-14-01427-t002:** Prevalence of overweight/obesity, anemia, micronutrient deficiency and co-occurrence of overweight/obesity with anemia or micronutrient deficiency among non-pregnant women 15–49 years of age in Ghana.

		Single Burden									Double Burden Malnutrition					
Characteristic		OWOB (A)		*p*-Value ^c^	Anemia (B)		*p*-Value	≥1 Micronutrient def. (C) ^d^		*p*-Value	A + B		*p*-Value	A + C		*p*-Value
		*n* = 1002			*n* = 999			*n* = 466			*N* = 989			*N* = 457		
Household		% ^a^	95% CI ^b^		% ^a^	95% CI ^b^		% ^a, d^	95% CI ^b^		% ^a^	95% CI ^b^		% ^a^	95% CI ^b^	
Stratum																
	Southern, %	47.2	(40.3, 54.2)	<0.001	24.0	(18.9, 29.9)	0.030	59.5	(48.8, 69.5)	0.725	8.3	(5.1, 13.4)	0.121	28.4	(20.9, 37.3)	0.029
	Middle, %	41.1	(34.5, 48.1)		17.4	(14.1, 21.4)		61.2	(52.9, 68.9)		6.9	(4.5, 10.5)		24.8	(18.1, 33.0)	
	Northern, %	18.5	(12.2, 26.9)		27.6	(20.4, 36.2)		65.8	(53.8, 76.0)		3.0	(1.4, 6.3)		11.8	(6.6, 20.3)	
Place																
	Urban, %	49.6	(43.3, 55.9)	<0.001	21.6	(17.5, 26.4)	0.964	63.8	(57.2, 70.0)	0.393	8.4	(3.1, 8.2)	0.107	29.5	(22.3, 37.9)	0.009
	Rural, %	28.8	(24.2, 34.0		21.8	(17.4, 26.9)		59.2	(50.2, 67.6)		5.0	(3.1, 8.2)		17.7	(13.3, 23.0)	
Wealth																
	Low, %	21.5	(16.4, 27.6)	<0.001	22.0	(16.7, 28.4)	0.213	56.3	(46.7, 65.5)	0.398	1.4	(0.5 3.6)	<0.001	13.2	(8.1, 20.8)	0.001
	Medium, %	39.6	(34.8, 44.7)		24.5	(19.3, 30.6)		64.4	(54.8, 73.0)		10.2	(6.9, 14.8)		21.4	(16.6, 27.2)	
	High, %	56.5	(49.9, 62.9)		18.0	(14.1, 22.7)		63.0	(54.5, 70.8)		8.0	(5.0, 12.6)		35.7	(25.3, 47.6)	
Sanitation																
	Unimproved	38.3	(34.1, 42.7)	0.407	20.9	(18.1, 24.1)	0.140	60.9	(54.5, 66.9)	0.528	5.7	(4.1, 7.9)	0.001	24.0	(19.4, 29.3)	0.622
	Improved	43.5	(31.1, 56.6)		26.8	(19.1, 36.2)		66.2	(50.7, 78.8)		13.1	(8.5, 19.8)		20.4	(10.0, 37.1)	
Water source																
	Unimproved	23.4	(17.0, 31.3)	0.001	17.2	(12.7, 23.0)	0.138	57.5	(39.5, 73.6)	0.628	4.2	(1.3, 12.8)	0.403	15.6	(7.7, 29.0)	0.194
	Improved	40.6	(35.8, 45.6)		22.2	(18.9, 25.8)		61.9	(56.0, 67.4)		6.9	(5.0, 9.5)		24.3	(19.5, 29.7)	
Women																
Women age																
	15-to 24, %	16.2	(12.0, 21.5)	<0.001	23.7	(18.1, 30.2)	0.179	66.5	(56.6, 75.2)	0.013	2.9	(1.6, 5.2)	<0.001	12.5	(7.3, 20.6)	0.002
	25-to 34, %	50.8	(43.8, 57.7)		23.2	(18.8, 28.2)		66.3	(57.2, 74.3)		10.2	(6.8, 15.1)		30.5	(22.9, 39.3)	
	35-to 49, %	56.9	(48.5, 65.0)		16.9	(12.3, 22.7)		49.6	(41.2, 58.1)		7.6	(4.8, 11.9)		29.2	(21.4, 38.6)	
Formal Education																
	None, %	34.2	(26.0, 43.5)	0.283	24.3	(16.9, 33.7)	0.468	61.3	(49.2, 72.2)	0.984	5.3	(3.0, 9.3)	0.407	20.8	(13.9, 29.8)	0.556
	Partly or fully literate, %	40.0	(34.8, 45.4)		21.1	(17.8, 24.8)		61.4	(55.3, 67.2)		7.0	(4.9, 9.9)		24.0	(18.5, 30.4)	
Dietary diversity																
	<5 food groups	40.2	(34.1, 46.6)	0.495	23.9	(19.4, 29.0)	0.118	63.1	(56.3, 69.5)	0.473	8.0	(5.4, 11.9)	0.100	24.1	(18.0, 31.4)	0.849
	≥5 food groups	37.6	(32.2, 43.4)		19.3	(15.8, 23.2)		59.7	(51.5, 67.4)		5.2	(3.4, 7.8)		23.0	(16.2, 31.7)	
	**Total**	**39.0**	**(34.5, 43.7)**		**21.7**	**(18.7, 25.1)**		**61.5**	**(55.8, 66.9)**		**6.7**	**(4.9, 9.1)**		**23.6**	**(19.1, 28.7)**	

^a^ Percentages. ^b^ CI = confidence interval, calculated considering the complex sampling design. ^c^ P-values measuring the differences in prevalence between household and women’s categories using chi-square test. ^d^ Prevalence of ≥1 micronutrient deficiency, including deficiencies in iron, vitamin A, folate, and vitamin B12.

**Table 3 nutrients-14-01427-t003:** Comparison of the observed co-occurrence of overweight/obesity with anemia and overweight/obesity with ≥1 micronutrient deficiency among non-pregnant women 15–49 years of age in Ghana with their expected prevalence.

Population	Observed Co-Occurrence (%)	Expected Co-Occurrence (%)	χ^2^	*p*-Value
**Southern**				
OWOB + anemia	8.3	11.3	0.008	0.928
OWOB + micronutrient def.	28.4	13.2	<0.001	1.000
**Middle**				
OWOB + anemia	6.9	7.2	<0.001	1.000
OWOB + micronutrient def.	24.8	25.2	<0.001	1.000
**Northern**				
OWOB + anemia	3.0	5.1	0.009	0.927
OWOB + micronutrient def.	11.8	12.2	<0.001	1.000
**Total**				
OWOB + anemia	6.7	8.5	0.004	0.951
OWOB + micronutrient def.	23.6	24.0	<0.001	0.992

OWOB represents overweight/obesity. Micronutrient deficiency represents ≥1 micronutrient deficiency (deficiencies in iron, vitamin A, folate, and vitamin B12).

**Table 4 nutrients-14-01427-t004:** Logit regression analysis of the correlates of the co-occurrence of overweight/obesity (OWOB) with anemia or ≥1 micronutrient deficiency among non-pregnant women 15–49 years of age in Ghana.

		OWOB			Anemia			≥1 Micronutrient Deficiency ^a^		
Household Characteristics		OR ^a^ (95% CI)	*p*-Value	dy/dx ^b^	OR (95% CI)	*p*-Value		OR (95% CI)	*p*-Value	dy/dx
Strata (ref: Southern)										
	Middle	0.9 (0.6, 1.3)	0.706	−0.012	**0.6 (0.4, 0.9)**	**0.011**	−0.079	1.3 (0.8, 2.2)	0.209	0.072
	Northern	**0.6 (0.3, 0.8)**	**0.007**	−0.110	1.0 (0.6, 1.5)	0.905	−0.005	**2.5 (1.4, 4.4)**	**0.002**	0.203
Place of residence (ref: Urban)										
	Rural	0.7 (0.5, 1.2)	0.212	−0.045	1.0 (0.6, 1.4)	0.931	−0.003	1.0 (0.5, 1.7)	0.990	−0.001
Wealth (ref: Low)										
	Medium	**1.6 (1.0, 2.4)**	**0.030**	0.084	1.3 (0.9, 2.0)	0.139	0.054	**2.0 (1.1, 3.5)**	**0.016**	0.156
	High	**2.9 (1.7, 5.1)**	**<0.001**	0.206	1.0 (0.6, 1.8)	0.756	0.014	1.7 (0.8, 3.5)	0.162	0.120
Sanitation (ref: unimproved)										
	Improved	0.8 (0.5, 1.2)	0.361	0.352	1.4 (0.9, 2.2)	0.110	0.065	1.1 (0.6, 2.1)	0.840	0.015
Water source (ref:unimproved)										
	Improved	1.2 (0.7, 1.9)	0.503	0.030	1.2 (0.8, 2.0)	0.269	0.042	1.1 (0.6, 2.0)	0.739	0.024
**Individual characteristics**										
	Age in years (ref: 15 to 24)									
	25 to 49	**5.5 (3.7, 8.0)**	**<0.001**	0.300	**0.7 (0.5, 0.9)**	**0.021**	−0.067	0.7 (0.4, 1.1)	0.150	−0.078
Formal education (ref: none)										
	Partly or fully literate	1.4 (0.9, 2.0)	0.111	0.056	0.8 (0.5, 1.1)	0.157	−0.048	0.9 (0.5, 1.5)	0.707	−0.022
Marital status (ref: Single)										
	Married/co-habiting	**1.8 (1.2, 2.6)**	**0.002**	0.103	1.1 (0.8, 1.5)	0.573	0.016	1.0 (0.6, 1.5)	0.935	−0.004
Dietary diversity (ref: <5 food groups consumed)										
	≥5 food groups	0.8 (0.6, 1.0)	0.079	−0.048	1.0 (0.7,1.3)	0.902	−0.003	0.8 (0.5, 1.2)	0.267	−0.051

^a^ Values represent adjusted odds ratios (95% Confident intervals) ^b^ ≥1 micronutrient deficiency (iron, vitamin A, folate, or vitamin B12). ^c^ dy/dx represents the change in the dependent variables (i.e., OWOB, anemia and micronutrient deficiency) for a unit change in independent variable.

## Data Availability

The data underlying the results presented in the study are owned by UNICEF Ghana and the Ministry of Health Ghana and contain confidential, identifying information. Data are available from UNICEF Ghana (accra@unicef.org) for researchers who meet the criteria for access to confidential data. The authors had no special access to the data.

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
