# Peer review of "Co-Occurrence of Overweight/Obesity, Anemia and Micronutrient Deficiencies among Non-Pregnant Women of Reproductive Age in Ghana: Results from a Nationally Representative Survey"

_nutrients, 2022, doi:10.3390/nu14071427_

Round 1
Reviewer 1 Report
Thank you for involving me in the review of this article. However, authors must carefully check if you have used people-first language with respect to the condition of obesity throughout the entire text. "Obese" is very inappropriate.
Author Response
Please the word "obese" in the text has be replaced with " respondents with obesity" throughout the text. Thank you
Reviewer 2 Report
Poorly written abstract with no conclusion and recommendation.
Line 20 incomplete
Addressing OWOB compared with anemia and MNDs in WRA in Ghana requires different interventions to alleviate obstetrics complications and infant mortality.
Line 59
The non-nutritional factors such as
Line 80
Personal interviews among selected subjects/women obtained information on age, educational status, and recent food supplement consumption.
Line 142-151
This must be included in Methods section rather than Results.
Start the results section from line 151.
The vast majority
Line 268
Author Response
- Poorly written abstract with no conclusion and recommendation.
We have revised the abstract for clarity and specificity and hope that this reviewer’s concerns are adequately addressed. If not, we would appreciate a more specific feedback on what should be changed further to fully satisfy this comment.
- Line 20 incomplete
Addressing OWOB compared with anemia and MNDs in WRA in Ghana requires different interventions to alleviate obstetrics complications and infant mortality.
Sentence revised
- Line 59
The non-nutritional factors such as
Sentence revised.
Line 80
- Personal interviews among selected subjects/women obtained information on age, educational status, and recent food supplement consumption.
This sentence has been modified for better clarity.
- Line 142-151This must be included in Methods section rather than Results.
This has been moved to the methods section as suggested.
- Start the results section from line 151.
The vast majority
Please have now stated the result from. The mean age of women………..(since that was part of information we obtained from analysis, thuis consider that as results. Thank you.
Line 268 This sentence has been modified for better clarity.
Reviewer 3 Report
The authors aimed to examine the prevalence of overweight/obesity, anemia, and micronutrient deficiencies and their co-occurrence and risk factors among non-pregnant women 15–49 years of age in Ghana. Data were from a 2017 two-stage national survey of 1063 women.
The manuscript it is relevant and interesting on an original topic. The paper well written, and the text clear and easy to read. The conclusions are consistent with the evidence and arguments presented and they address the main question posed.
I congratulate the authors for the manuscript and I guess it could be of great interest for the readers of the Nutrients Journal.
Author Response
Thank you for the encouraging comments.
Reviewer 4 Report
Line 7 --> Add affiliations
Line 145 --> Why did only 50% tested for Folate and B12?
Line 150 --> should be “were 15- 24 y of age (38%) and were married (60%)
Line 175 --> Should be space between the end of the Table legend and the start of the next paragraph
Table 2 --> make sure the word p-value next to OWOB is the same font and style as the others. Also were there any women who had all 3 (i.e., OWOB, anemia and micronutrient deficiency)?
Table 3 --> Do you have confidence intervals for these values? Also, I would recommend having the word “co-occurrence” on the same line if possible
Table 4 --> Under place of residence, you only list the information for rural, what about urban? Also, there should be a space between rural and wealth since Wealth is separate category
Line 256 --> Weird wording
Line 263 --> remove extra space between “wealth” and “and overweight/obesity”
Line 296 --> make sure to define SSA
Line 307/308 --> Why wasn’t Zinc examined?
Line 321 --> Obesity has long been causally linked with chronic low-grade inflammation so you don’t need to assume
Line 327 --> should be joined with the previous line instead of in a new paragraph
Author Response
- Why did only 50% tested for folate and B12?
This decision was made based on two criteria, one being the relatively high cost of laboratory analysis for these two biomarkers, the other being the fact that national data did not exist previously at all for Ghana. Therefore, national stakeholders opted for an approach that would provide sufficient precision for a national estimate to understand whether or not folate/B12 deficiencies are of public health concern. Using a random sub-sample provided this information at the very least at considerable cost saving.
- Line 150 --> should be “were 15- 24 y of age (38%) and were married (60%)
This has been corrected
- Line 175 --> Should be space between the end of the Table legend and the start of the next paragraph
This has been corrected
- Table 2 --> make sure the word p-value next to OWOB is the same font and style as the others. Also were there any women who had all 3 (i.e., OWOB, anemia and micronutrient deficiency)?
Fonts and style differences have been resolved. Please we did not look at women who had all 3 malnutrition conditions. That was not our objective.
- Table 3 --> Do you have confidence intervals for these values? Also, I would recommend having the word “co-occurrence” on the same line if possible
“co-occurrence” is now on the same line. The expected frequencies were derived/calculated from the mean values obtained from the observed single malnutrition conditions. Although we could have generated confidence individuals for the observed values we could not have that for the expected, thus for consistency, we decided to leave out the confident intervals in this table.
- Table 4 --> Under place of residence, you only list the information for rural, what about urban? Also, there should be a space between rural and wealth since Wealth is separate category
This table represents our regression analysis. Urban residence is indicated as the comparison group (i.e., reference) for the rural location. A line has been introduced to separate the wealth as suggested.
- Line 256 --> Weird wording
The sentence has been reworded for clarity.
- Line 263 --> remove extra space between “wealth” and “and overweight/obesity”
Please this has now been done
- Line 296 --> make sure to define SSA
Done
- Why wasn't Zinc examined in the study?
Zinc was not included due to cost considerations and logistic challenges in order to obtain valid samples (dust free environment to avoid contamination, something which was difficult in certain areas of Ghana, where the teams had to operate using mobile laboratories due to lack of existing laboratory infrastructure). But also, existing literature indicated that for Ghana, Zinc deficiency may be less of a public health issue than one would think: Lartey et al. found in a cohort of young children, that zinc deficiency was present in about 5% of these at-risk children [1] and a more recent study based on dietary zinc intake estimates by Beal et al. reported a low risk for zinc deficiency in Ghana [2].
- Line 321 --> Obesity has long been causally linked with chronic low-grade inflammation, so you don’t need to assume
This is corrected.
- Line 327 --> should be joined with the previous line instead of in a new paragraph
Done